# Unforgotten Safety: Preserving Safety Alignment of Large Language Models with Continual Learning

## Abstract

The safety alignment of large language models (LLMs) is becoming increasingly important with their democratization. In this paper, we study the safety degradation that comes with adapting LLMs to new tasks. We attribute this safety compromise to catastrophic forgetting and frame the problem of preserving safety when fine-tuning as a continual learning (CL) problem. We consider the fine-tuning-as-a-service setup where users upload their data to a service provider to get a customized model that excels on the user's selected task. We adapt several CL approaches from the literature and systematically evaluate their ability to mitigate safety degradation. These include regularization-based, memory-based, and model merging approaches. We consider two scenarios, (1) benign user data and (2) poisoned user data. Our results demonstrate that CL approaches consistently achieve lower attack success rates than standard fine-tuning. Among these, DER outperforms both other CL methods and existing safety-preserving baselines while maintaining task utility. These findings generalize across three downstream tasks (GSM8K, SST2, Code) and three model families (LLaMA2-7B, Mistral-7B, Gemma-2B), establishing CL as a practical solution to preserve safety.

## 1 Introduction

Large language models (LLMs) have achieved remarkable capabilities across different domains. However, capability alone without rigorous safety alignment is insufficient for responsible deployment. To address LLM safety, models are aligned prior to deployment using supervised fine-tuning (SFT), reinforcement learning from human feedback (RLHF) Christiano et al. (2017), or direct preference optimization (DPO) Rafailov et al. (2023). However, subsequent fine-tuning of an aligned model on additional data can degrade its safety, causing the model to forget or override safe behavior Qi et al. (2023); Lermen et al. (2023). As an emerging paradigm, LLM fine-tuning services enable end users to upload their own data for fine-tuning. Fine-tuning can erode the already learned safe behavior, even when the data is entirely benign. Moreover, a few harmful samples mixed into benign data can drastically compromise alignment. Consequently, there is an urgent need for an effective mitigation strategy since the providers of these services are responsible for the output of the fine-tuned models.

In this context, we observe that safety degradation during benign fine-tuning closely mirrors catastrophic forgetting as mentioned in Qi et al. (2023), a well-studied phenomenon in continual learning (CL). The model's safe behavior fades with subsequent adaptation to new tasks. Furthermore, a small amount of harmful content in the user data can produce a model that easily complies with harmful prompts. Motivated by this, we frame safety-preserving fine-tuning as a CL problem. We consider two scenarios: (1) benign fine-tuning, where user data contains no harmful content, and (2) poisoned fine-tuning, where user data contains harmful samples. When the data is poisoned, the threat is no longer catastrophic forgetting only, but also the presence of conflicting samples with what was previously learned.

We adapt several CL approaches for safety-preserving fine-tuning from different CL paradigms, including memory-based (A-GEM, DER, Refresh Learning), regularization-based (LwF, EWC), and model merging approaches (MagMax). We systematically evaluate these methods across three downstream tasks (GSM8K,

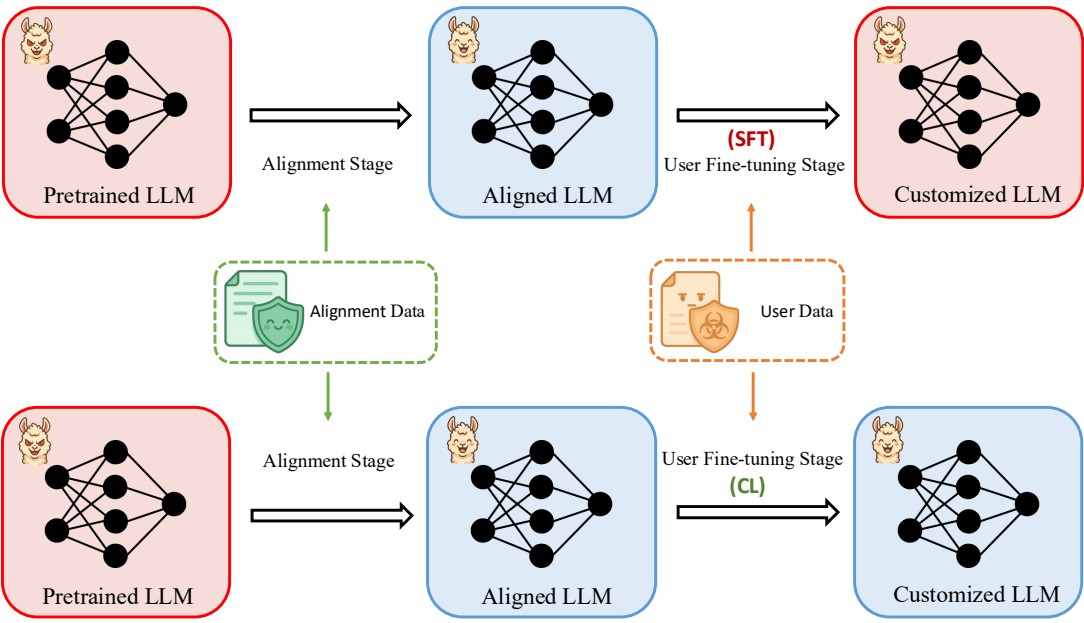

Figure 1: **Safety Preservation in Fine-tuning-as-a-Service.** A pretrained LLM undergoes alignment to become safe. Standard fine-tuning (SFT, top) causes catastrophic forgetting of safety alignment, reverting the model to unsafe behavior—this degradation worsens when user data is poisoned (contains harmful samples). Our adapted continual learning approaches (CL, bottom) preserves safety whether user data is benign or poisoned.

SST2, Code) under both benign and poisoned fine-tuning scenarios. Our comprehensive evaluation reveals that continual learning methods effectively reduce safety degradation compared to standard fine-tuning. Among these approaches, DER achieves the best balance of safety and utility across benign and poisoned fine-tuning scenarios, outperforming both other CL methods and existing safety-preserving baselines.

Our main contributions are:

- We frame safety-preserving fine-tuning as a CL problem and comprehensively evaluate CL approaches against existing safety-preserving baselines.

- We demonstrate that CL methods effectively preserve safety alignment in both benign and poisoned fine-tuning scenarios, with DER achieving optimal balance of safety and utility.

- We validate our findings across three tasks and three model families, establishing the generalizability of CL for safety-preserving fine-tuning.

## 2  Related Work

**Finetuning Attacks.**  Recent studies have demonstrated that the process of fine–tuning large language models (LLMs) introduces an exploitable attack surface that adversaries can leverage to compromise safety alignment. For example, Yang et al. (2023) introduced the concept of 'Shadow Alignment', in which even a small set (around 100) of deliberately malicious examples inserted during fine–tuning can cause the model's safety constraints to break down. Similarly, Qi et al. (2023) showed that even benign fine–tuning data may inadvertently shift a model's behavior toward harmful outputs, highlighting an attack vector where

seemingly innocuous data undermine safety. In addition, Yi et al. (2024a) demonstrated that both standard supervised fine–tuning and preference–based optimization (e.g., RLHF) can be subverted when harmful data are introduced. In the parameter–efficient regime, Lermen et al. (2023) showed that methods like LoRA are not immune—malicious fine–tuning using such techniques can also erode safety. More covert attack strategies have emerged as well: for example, Halawi et al. (2024) proposed a "Covert Malicious Fine–tuning" approach that encodes harmful data to bypass moderation filters, while Chen et al. (2024) and Hawkins et al. (2024) have explored model editing and developer–tuned scenarios. Finally, Poppi et al. (2024) extended the discussion by demonstrating that even a few harmful examples in one language can generalize to compromise multilingual LLMs.

**Finetuning Defenses.** In response to these vulnerabilities, a diverse array of defensive strategies have been proposed. Several approaches target the initial alignment stage; for example, Huang et al. (2024d) introduces Vaccine, which uses perturbation–aware optimization to "vaccinate" the model against subsequent harmful fine–tuning, while Rosati et al. (2024) proposes RepNoise that incorporates harmful data with added noise to actively repel unsafe gradients. Other methods, such as CTRL from Liu et al. (2024), rely on data–centric techniques by blending curated, safe data into the alignment process, whereas Liu et al. (2025) modifies training data by previewing partial answer prefixes. Tamirisa et al. (2024) (TAR) and Huang et al. (2024c) (Booster) use meta–learning and gradient–regularization techniques to simulate and counteract harmful fine–tuning perturbations. At the fine–tuning stage itself, defenses such as LDIFS Mukhoti et al. (2023) employ distance regularization (e.g. via KL–divergence) to constrain model drift, and SafeInstr Bianchi et al. (2024) continuously reinforces safety by mixing alignment data during fine–tuning. Furthermore, post–fine-tuning repair methods like LAT Casper et al. (2024), SOMF Yi et al. (2024b), Antidote Huang et al. (2024a), and SafetyLock Zhu et al. (2024) have been developed to re–align models after harmful fine–tuning has occurred. Together, these defenses form a multi–stage toolkit that seeks to preserve safety while allowing task–specific customization—even if sometimes at the expense of increased computational overhead or complexity.

**Continual Learning.** aims to develop models that can learn from sequence of tasks with minimal forgetting Rolnick et al. (2019). Memory-based approaches mitigate forgetting storing samples from previous task in a fixed size buffer Rolnick et al. (2019); Buzzega et al. (2020b); Rebuffi et al. (2017), and they yield to better performance compared to regularization-based approaches. Regularization-based methods aim to preserve past knowledge by constraining weight updates, typically by identifying the importance of parameters, like Elastic Weight Consolidation (EWC) Kirkpatrick et al. (2017) and Synaptic Intelligence (SI) Zenke et al. (2017). GEM Lopez-Paz & Ranzato (2017) and its more efficient variant A-GEM Chaudhry et al. (2018) optimize gradient-based episodic memory to mitigate forgetting. On the other hand, Refresh Learning Wang et al. (2024) address the over-memorization phenomenon by introducing unlearning step prior to optimizing for a given batch.

In this work, we establish a connection between safety alignment and continual learning, framing the loss of alignment during fine-tuning as a catastrophic forgetting problem. We demonstrate that existing continual learning methods can effectively mitigate this alignment degradation.

## 3 Preliminaries

In this section, we introduce the core concepts and notation used throughout the paper: (i) we define the setup targeted in this work, (ii) we formalize safety alignment in LLMs, and finally (iii) we present the standard CL framework to mitigate catastrophic forgetting.

### 3.1 Threat Model

We adopt a two-stage pipeline, also referred to as Fine-tuning-as-a-Service. The service provider first aligns a pretrained base model using their curated safety dataset $\mathcal{D}_{\text{safe}}$. Then a user uploads their data $\mathcal{D}_{\text{user}}$ of $n$ examples for fine-tuning. The service provider controls the safety alignment and the fine-tuning procedure, but not the user's uploaded data. The user's data may contain harmful samples. We denote the proportion

of the harmful samples to the total fine-tuning dataset by poison ratio $p$. Our objective is to adapt the model on $\mathcal{D}_{\text{user}}$ while preventing safety degradation even when $p > 0$. After these two stages, the resulting model is hosted by the provider and the user sends request to it in order to fulfill their task.

## 3.2   Safety Alignment of LLMs

We define safety alignment as the property that a language model consistently respects ethical and policy-based constraints. In particular, we expect the LLM to comply with regular safe prompts and refuse to respond to harmful prompts. Formally, let $\mathcal{X}_{\text{safe}}$ be a set of prompts (or instructions) that are either unsafe (e.g., requesting harmful actions) or require cautious handling, and let $\mathcal{Y}_{\text{safe}}$ be their intended safe responses (often refusals or redirections). A model is deemed safety-aligned if for every $(x, y) \in \mathcal{D}_{\text{safe}} \subseteq (\mathcal{X}_{\text{safe}}, \mathcal{Y}_{\text{safe}})$, the model's output is safe according to predefined policies or human feedback.

In practice, $\mathcal{D}_{\text{safe}}$ is curated from alignment data (e.g., red-teaming prompts, refusal exemplars, policy-driven dialogues) and used to fine-tune a base model via supervised training or reinforcement learning from human feedback (RLHF). However, subsequent fine-tuning of the model, even on some new benign dataset, can lead to safety degradation. In the context of CL, the model can potentially forget its safe behavior due to perturbing parameters crucial for safety.

## 3.3   Continual Learning

Continual Learning (CL) studies the problem of training a single model on a sequence of tasks $\{T_1, T_2, \ldots, T_n\}$ without compromising performance on previously learned tasks Parisi et al. (2019). A central challenge in CL is catastrophic forgetting, which occurs when the model's parameters adapted for a new task $T_k$ interfere with the knowledge acquired for older tasks $T_1, \ldots, T_{k-1}$.

In the context of model customization and fine-tuning, we view safety-alignment task on $\mathcal{D}_{\text{safe}}$ as $T_1$ and domain-specific fine-tuning on user's data $\mathcal{D}_{\text{user}}$ as $T_2$. Let $\theta_0$ denote the pretrained base model (before safety alignment), and let $\theta_{\text{safe}}$ be the safety-aligned model obtained by training the base model on $\mathcal{D}_{\text{safe}}$. Our objective is to update the model parameters from $\theta_{\text{safe}}$ to $\theta^*$ using $\mathcal{D}_{\text{user}}$, while retaining strong performance on the safety task. In other words, if $\mathcal{L}_{\text{safe}}(\theta_{\text{safe}})$ is the (supervised) loss on $\mathcal{D}_{\text{safe}}$ after safety alignment, we want $\mathcal{L}_{\text{safe}}(\theta^*) \approx \mathcal{L}_{\text{safe}}(\theta_{\text{safe}})$. Naive fine-tuning only minimizes $\mathcal{L}_{\text{user}}$ on $\mathcal{D}_{\text{user}}$, often causing a substantial rise in $\mathcal{L}_{\text{safe}}$ and, consequently, a large drop in alignment.

CL algorithms introduce additional constraints or memory components to preserve performance on old tasks. In our formulation, the "old task" is the safety alignment.

# 4   Mitigating Safety Degradation

Prior work Qi et al. (2023); He et al. (2024) shows that fine-tuning a safety-aligned language model on a benign dataset, with no explicit harmful content, can degrade the model's safety. Figure 2 illustrates the drop in safety, measured by increased Attack Success Rate (ASR), after fine-tuning LLaMA2-7B[1] and Mistral-7B[2] models on GSM8K, SST2, and Code[3] datasets. The problem becomes more severe when the fine-tuning dataset is poisoned with harmful content. Viewing this safety degradation as catastrophic forgetting, we investigate how CL approaches can preserve safety alignment when fine-tuning on both benign and poisoned data. Building on the formulation in section 3.3, we adapt several CL approaches for fine-tuning LLMs while minimizing safety degradation compared to standard fine-tuning (SFT).

## 4.1   Continual Learning Approaches

We adapt several continual learning approaches to preserve safety alignment when fine-tuning on both benign and poisoned data. These methods can be categorized into three main families: regularization-based

---

[1]https://huggingface.co/meta-llama/Llama-2-7b-hf
[2]https://huggingface.co/mistralai/Mistral-7B-v0.3
[3]https://huggingface.co/datasets/nvidia/Nemotron-Post-Training-Dataset-v1

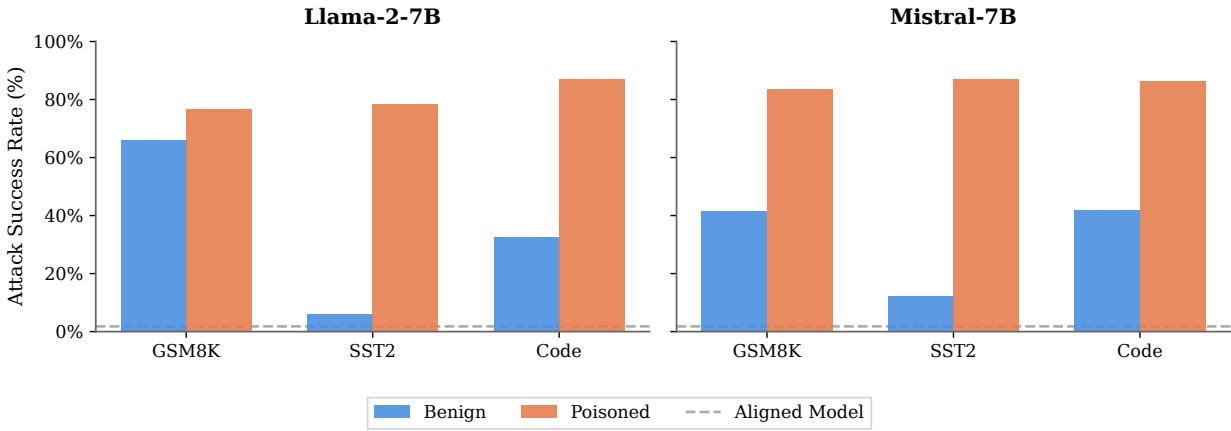

Figure 2: **Safety Degradation After Fine-tuning.** Attack Success Rate (ASR) increases after fine-tuning LLaMA-2 and Mistral on benign datasets (GSM8K, SST2, Code), with further degradation under poisoned data setup (contains harmful samples).

approaches that constrain parameter updates, memory-based approaches that leverage stored safety data, and model merging approaches that consolidate knowledge post-hoc. Below, we describe each adapted method and its mechanism for preserving safety.

### 4.1.1 Regularization-Based Approaches

These methods preserve prior knowledge by constraining how model parameters can change during fine-tuning, without requiring storage of previous task data.

**Elastic Weight Consolidation (EWC)** regularizes the parameter update during fine-tuning on a new task to prevent catastrophic forgetting by constraining changes to important weights Kirkpatrick et al. (2017). The key idea behind EWC is to estimate the importance of each parameter for previously learned tasks using the Fisher Information Matrix and penalize deviations from their learned values. It does that by minimizing:

$$\mathcal{L}(\theta) = \mathbb{E}_{(x,y)\sim\mathcal{D}_{\text{user}}}\big[\text{CE}(f_\theta(x), y)\big] + \sum_i \frac{\lambda}{2} F_i \left(\theta_i - \theta_i^{\text{safe}}\right)^2,$$

where $\text{CE}(\cdot,\cdot)$ denotes the cross-entropy loss on downstream task, $F$ is the Fisher Information Matrix, which quantifies the importance of each parameter based on how sensitive the loss function is to changes in that parameter. $\lambda$ is a hyper-parameter balancing the relative importance of the old task, safety alignmnt, with respect to the current task. Like other regularization methods, EWC must balance preserving prior knowledge and adapting to new tasks, which can be challenging.

**Learning without Forgetting (LwF)** preserves prior knowledge by distilling the predictions of the safety-aligned teacher while training on the downstream task Li & Hoiem (2017). In our setup, we distill the last hidden states denoted as $h_\theta(x)$, $h_{\theta_{\text{safe}}}(x) \in \mathbb{R}^d$. Fine-tuning matches these representations on downstream task data, without storing data samples from the previous safety alignment task:

$$\mathcal{L}_{\text{LwF}} = \mathbb{E}_{(x,y)\sim\mathcal{D}_{\text{user}}}\big[\text{CE}(f_\theta(x), y)\big] + \beta\, \mathbb{E}_{x\sim\mathcal{D}_{\text{user}}}\big[\|h_\theta(x) - h_{\theta^{\text{safe}}}(x)\|_2^2\big],$$

where $\text{CE}(\cdot,\cdot)$ denotes the cross-entropy loss on downstream task samples and $\big[\|h_\theta(x) - h_{\theta^{\text{safe}}}(x)\|_2^2\big]$ is the squared error between the last hidden state produced by the current model ($h_\theta(x)$) and the ones produced by the safety-aligned model ($h_{\theta^{\text{safe}}}(x)$). The key advantage of LwF is its use of knowledge distillation rather

than parameter constraints, allowing more flexible adaptation to downstream tasks Oren & Wolf (2021).

### 4.1.2 Memory-Based Approaches

Unlike regularization approaches, memory-based methods maintain explicit access to safety data during fine-tuning, enabling more direct preservation of safety-aligned behavior.

**Average Gradient Episodic Memory (A-GEM)** stores $N$ samples from previous task, safety alignment, in episodic memory $\mathcal{B}$.

However, instead of replaying these samples directly, it uses gradient-based constraints to ensure that updates on the downstream task do not increase loss on stored safety samples Chaudhry et al. (2018). Specifically, if the dot product between the current task gradient $\mathbf{g}_{user}$ and the reference gradient computed on the buffer $\mathbf{g}_{safe}$ is negative, the current gradient is projected as follows:

$$\mathbf{g}_{user} = \mathbf{g}_{user} - \frac{\mathbf{g}_{user}^{\top}\mathbf{g}_{safe}}{\|\mathbf{g}_{safe}\|^2}\mathbf{g}_{safe}$$

This ensures that the loss on the stored safety samples does not increase, formally ensuring $\mathcal{L}_{\text{safe}}(\theta) \leq \mathcal{L}_{\text{safe}}(\theta_{\text{safe}})$. However, this mechanism may struggle when task gradients point in conflicting directions Yu et al. (2020).

**Dark Experience Replay (DER)** preserves previous knowledge by maintaining a fixed-size episodic memory $\mathcal{B}$ that stores not only safety data samples but also the corresponding logits produced by the safety-aligned model Buzzega et al. (2020a). Specifically, the buffer stores $N$ triples $(x, y, z)$ from $\mathcal{D}_{\text{safe}}$, where $x$ is an unsafe prompt, $y$ is its corresponding refusal response, and $z$ represents the pre-softmax logits of the safety-aligned model:

$$\mathcal{B} = \{(x_k, y_k, z_k)\}_{k=1}^N.$$

During fine-tuning on the downstream data, $\mathcal{D}_{\text{user}}$, DER replays samples from $\mathcal{B}$ and minimizes:

$$\mathcal{L}_{\text{DER}} = \mathbb{E}_{(x,y)\sim\mathcal{D}_{\text{user}}}\big[\text{CE}(f_\theta(x), y)\big] + \lambda\,\mathbb{E}_{(x,z)\sim\mathcal{B}}\big[\|f_\theta(x) - z\|_2^2\big],$$

where $\text{CE}(\cdot, \cdot)$ denotes the cross-entropy loss on downstream data, and $\|f_\theta(x) - z\|_2^2$ is the squared error between the current model's logits $f_\theta(x)$ and the stored logits $z$, produced by the safety-aligned model before fine-tuning. The scalar $\lambda > 0$ balances the two terms. By combining data replay with knowledge distillation on stored logits, DER provides a more robust mechanism for preserving safety compared to replay or distillation alone.

**Refresh Learning** addresses over-memorization of old tasks, which can hinder generalization Wang et al. (2024). Similar to other memory-based approaches Rolnick et al. (2019), our adaptation of Refresh Learning maintains a fixed-size episodic buffer $\mathcal{B}$ with $N$ pairs $(x, y)$ from $\mathcal{D}_{\text{safe}}$, where $x$ is an unsafe prompt and $y$ is its corresponding refusal response. During training on the downstream task, the method replays buffer samples by drawing from both $\mathcal{D}_{\text{user}}$ and $\mathcal{B}$ at every iteration, optimizing $L_{\text{user}}$, which is a cross-entropy loss on $\mathcal{D}_{\text{user}}$. It applies two steps at every mini-batch: an unlearning step to selectively forget unimportant information stored in the model weights, followed by a relearning step. The unlearning step is performed via the following weight update:

$$\theta^i = \theta^{i-1} + \gamma\big[F^{-1}\nabla L_{\text{user}}(\theta^{i-1})\big] + \mathcal{N}\big(0, 2\gamma F^{-1}\big),$$

where $F$ is the Fisher Information Matrix on the safety alignment task and $\gamma$ is the unlearning rate. To add some stochasticity to the unlearning step, a random noise $\mathcal{N}\big(0, 2\gamma F^{-1}\big)$ is introduced in the weight update

equation above.

### 4.1.3 Model Merging

Unlike regularization and memory-based methods that preserve knowledge during fine-tuning, model merging approaches operate post-hoc by combining models trained on different tasks.

**MagMax** enables continual adaptation of large pre-trained models to new tasks while mitigating catastrophic forgetting Marczak et al. (2024). This is accomplished through the combination of sequential fine-tuning on the given tasks and model merging. Specifically, after fine-tuning sequentially on the given tasks, it merges the models obtained from all seen tasks. The merging is done by calculating task vectors for all tasks:

$$\Delta_t = \theta_t - \theta_0,$$

where $\theta_0$ are the base model weights and $\theta_t$ the weights after fine-tuning on task $t$. In our scenario, we use two tasks, safety-alignment and finetuning on the downstream task. During merging, for each parameter index $i$ we select the element with the largest magnitude across tasks:

$$t^\star = \arg\max_t |\Delta_{t,i}|$$

Lastly, $t^\star$ is multiplied by the scaling factor $\lambda$ and added to the weights of the base model $\theta_0$.

Having described these diverse CL approaches, we now turn to their empirical evaluation. The following section assesses how effectively each approach preserves safety alignment while maintaining downstream task performance.

## 5 Experiments

We consider the fine-tuning-as-a-service setup in which a curated dataset is first used to safety-align the base model. The safety-aligned model is then fine-tuned on the downstream task provided by the user. We experiment with fully benign fine-tuning and poisoned fine-tuning scenarios. In the poisoned scenario, the task data includes vanilla harmful samples that can break the model's safety alignment. Given the sequential nature of the setup, it can be casted as a CL problem with two tasks. The first task is safety alignment, and the second task is the downstream fine-tuning task. In the poisoned scenario, the goal is to mitigate forgetting while minimizing the effect of the harmful prompts on the model's safety. In both scenarios, the end goal is a model that balances safety and utility.

### 5.1 Datasets

**Safety Alignment Dataset.** We utilize the vanilla subset of Wildjailbreak[4] dataset Jiang et al. (2024). It consists of harmful and benign prompts. Harmful prompts try to elicit harmful responses from the LLM deliberately, and they are coupled with detailed refusal responses. Benign prompts are seemingly harmful prompts that superficially resemble harmful prompts by keywords or discuss sensitive topics in a harmless manner. Including such benign safety samples is especially useful for combating the exaggerated safety behavior in LLMs, also known as over-refusal phenomenon. We randomly sample $5,000$ benign and $5,000$ harmful prompts for a total of $10,000$ samples for safety alignment.

**Downstream Task Datasets.** We consider three downstream skill acquisition tasks: sentiment analysis, math problem solving, and python code generation. For sentiment analysis, we use the SST2 dataset that presents a binary classification problem on single sentences from movie reviews Socher et al. (2013). For math problem solving, we consider GSM8K, which is a collection of easily verifiable math problems that require multi-step reasoning Cobbe et al. (2021). For python code generation, we use NVIDIA's NemotronCode,

---

[4]https://huggingface.co/datasets/allenai/wildjailbreak

a task oriented python code generation dataset with explanations Bercovich et al. (2025). We randomly sample $5,000$ samples from SST2 and GSM8K following previous literature Huang et al. (2024b) and $10,000$ samples from NemotronCode, as it is a harder skill to acquire.

**Poison Dataset.** We consider BeaverTails[5] dangerous subset for harmful prompts and responses for the poisoned fine-tuning scenario Ji et al. (2023). Following previous literature Huang et al. (2024b), we consider a default poison ratio of $p = 0.1$ unless stated otherwise. We also provide a robustness analysis at reasonably higher poison ratios in the ablation section.

## 5.2 Training Details and Hyperparameters

We use standard LoRA fine-tuning Hu et al. (2022) for its compute efficiency with rank $r = 8$, scaling factor $\alpha = 4$, and dropout of 0.1. We fix the training hyper-parameters across tasks and models. We train for a total of 3 epochs with a batch size of 5. We use a peak learning rate of $5e^{-5}$ and a cosine learning rate scheduler with 10% warm-up of all training steps. We employ a weight decay of 0.1. We showcase results on three different models that span three different model families and two sizes to show generalization of our observations and results. We choose LLaMA2-7B, Mistral-7B, and Gemma-2B[6].

## 5.3 Evaluation Details and Metrics

We report average Attack Success Rate (ASR%) on three different safety datasets: AdvBench Zou et al. (2023), DirectHarm Lyu et al. (2024), and HexPhi Qi et al. (2024) (all tables present the ASR averaged over these 3 datasets). For reproducibility, we resort to using `Llama-Guard-3-8B` Dubey et al. (2024), an open-source safety judge of the safety of the responses of our fine-tuned models. We also report a measure of utility for the three different tasks: classification accuracy on 1000 samples from the validation set of SST2, and zero-shot pass@1 on 500 samples from GSM8K test set, and zero-shot pass@1 on HumanEval/HumanEval+ test set (all tables present utility for the code task using two numbers: HumanEval score and HumanEval+ score (in parentheses)).

## 5.4 Main Results

This section investigates how adapting CL approaches for fine-tuning LLMs affects their safety alignment in the benign fine-tuning setting. Then we highlight the best-performing CL approaches and extend them to fine-tuning on poisoned data scenario, which is a more challenging setup, and compare against three baselines from the literature, SafeInstr Bianchi et al. (2024), SafeLoRA Hsu et al. (2024), and Lisa Huang et al. (2024b).

### 5.4.1 Fine-tuning on Benign Data

Viewing the degradation of model's safety as a catastrophic forgetting problem, we investigate to what extent safety can be preserved by adapting different continual learning (CL) approaches to benign fine-tuning. Table 1 presents the results of fine-tuning the safety-aligned LLaMA2-7B model on GSM8K, SST2, and Code datasets. SFT exhibits large safety degradation, with particularly high ASR on GSM8K (66.0%) and Code (46.4%). In contrast, adapted CL approaches drastically reduce ASR across all datasets, confirming that continual learning methods can effectively mitigate safety forgetting.

**Memory-Based Methods.** DER and A-GEM show strong safety preservation with an ASR $< 2\%$ and minimal utility loss across all datasets (less than 2 percentage points). Refresh Learning demonstrates exceptional safety performance with ASR below 1%, though this comes at the cost of significant utility degradation on SST2 (a drop of 8.1%). Notably, Refresh maintains excellent utility on the Code task (20.1% vs. 20.7% for SFT), which might indicate sensitivity to the downstream task.

**Regularization Methods.** EWC maintains relatively high ASR on GSM8K (44.8%) and suffers substantial utility degradation on the Code task (8.5% compared to SFT's 20.7%). This severe utility drop is not

---

[5]https://huggingface.co/datasets/PKU-Alignment/BeaverTails
[6]https://huggingface.co/google/gemma-2-2b

| Method | GSM8K | | SST2 | | Code | |
|---|---|---|---|---|---|---|
| | ASR% ↓ | Pass@1 ↑ | ASR% ↓ | Acc ↑ | ASR% ↓ | Pass@1 ↑ |
| SFT | 66.0 | 24.8 | 6.2 | 95.4 | 46.4 | 20.7(17.7) |
| DER | 0.5 | 22 | 0.7 | 94.4 | 0.5 | 18.9 (16.5) |
| Refresh | **0.2** | 22.0 | **0.2** | 87.3 | **0.0** | **20.1 (17.7)** |
| A-GEM | 0.9 | 22.6 | 1.6 | **95.4** | 0.4 | **20.7(17.1)** |
| EWC | 44.8 | **23.8** | 1.6 | 92.2 | 8.7 | 8.5 (6.7) |
| LwF | 2.7 | 20.4 | 2.1 | 94.5 | 4.3 | 18.9 (15.2) |
| MagMax | 16.0 | 22.8 | 2.2 | 90.4 | 9.3 | 14.6 (14.0) |

Table 1: **CL Approaches Mitigate Safety Forgetting.** This table shows that adapting CL approaches for fine-tuning the safety-aligned model LLaMA2-7B on the benign downstream task reduces the average attack success rate (ASR%) compared to SFT. We report the average ASR% across three datasets and utility on the corresponding downstream task.

surprising; EWC is known to face challenges in achieving an optimal stability-plasticity balance Kirkpatrick et al. (2017). On the other hand, LwF demonstrates more balanced performance with moderate ASR ($\leq 4.3\%$) and reasonable utility across all tasks. This consistency comes from LwF's softer regularization approach through knowledge distillation, which is more flexible than hard parameter constraints Oren & Wolf (2021).

**Model Merging.** MagMax reduces ASR compared to SFT but fails to match memory-based methods while exhibiting poor Code utility (14.6%). This suggests that model merging by selecting or interpolating weights after training, may not optimally balance conflicting objectives such as safety preservation and downstream task performance.

Overall, DER, A-GEM, and LwF exhibit consistent behavior across tasks with a reasonable safety–utility trade-off. Accordingly, we carry forward these three methods and evaluate them under the more challenging poisoned data setting.

### 5.5 Fine-tuning on Poisoned Data

We examine the more challenging scenario where downstream training data contains harmful content at poison ratio $p = 0.1$. We evaluate the CL approaches that achieved low ASR with balanced trade-off between safety and utility in the benign setting (DER, A-GEM, LwF) alongside three safety-preserving baselines: SafeInstr, Lisa, and SafeLoRA. Table 2 presents the results.

**CL Methods.** DER and LwF achieve low ASR, ($\leq 5.9\%$ and $\leq 4.3\%$, respectively), with a utility close to SFT. In contrast, A-GEM exhibits significant degradation with ASR rising to (6.8-16.6%). This aligns with prior work showing that gradient projection methods struggle when task objectives conflict Yu et al. (2020), as safety alignment and harmful content represent contradictory objectives.

**Baselines.** Lisa achieves the lowest ASR (1.2-6.6%) with competitive utility. SafeInstr demonstrates moderate ASR (5.8-7.2%), while SafeLoRA remains relatively unsafe ($\geq 24.4\%$ ASR).

Based on these findings, we carry forward DER, LwF, and Lisa for detailed ablation studies. These three methods have demonstrated a consistent and effective safety-utility trade-off in both benign and poisoned settings.

### 5.6 Ablations

In this subsection, we evaluate the robustness and reliability of the top-performing methods. First, we investigate their robustness to increasing poison ratios. Second, we assess their susceptibility to exaggerated safety (over-refusal). Finally, for methods that use safety data during fine-tuning, we examine the impact of reducing the number of safety samples.

| Method | GSM8K | | SST2 | | Code | |
|---|---|---|---|---|---|---|
| | ASR% ↓ | Pass@1 ↑ | ASR% ↓ | Acc ↑ | ASR% ↓ | Pass@1 ↑ |
| SFT | 76.7 | 22.4 | 78.3 | 94.5 | 86.9 | 20.1 (15.9) |
| SafeInstr | 5.8 | 21.4 | 6.4 | 94.7 | 7.2 | **19.5 (16.5)** |
| Lisa | **1.8** | 20.0 | 6.6 | **95.1** | **1.2** | 17.1 (15.2) |
| SafeLoRA | 24.4 | **23.0** | 72.4 | 94.7 | 35.5 | 18.3 (16.5) |
| DER | 2.3 | 20.8 | 5.9 | 94.3 | 1.7 | **19.5 (16.5)** |
| A-GEM | 16.6 | 20.6 | 15.0 | 94.6 | 6.8 | 17.7 (15.2) |
| LwF | 2.7 | 20.4 | **2.1** | 94.5 | 4.3 | 18.3 (14.6) |

Table 2: **CL Approaches for Preserving Safety in Poisoned Data Setup.** This table shows that CL adaptations of safety-aligned LLaMA2-7B reduce ASR% compared to SFT for poisoned fine-tuning ($p = 0.1$). Some CL approaches show better or comparable performance to the reported baselines. We report the average ASR% across three datasets and utility on the corresponding downstream task.

| Methods | ASR% ↓ | | | | Pass@1 ↑ | | | |
|---|---|---|---|---|---|---|---|---|
| | clean | $p$=0.1 | $p$=0.2 | $p$=0.3 | clean | $p$=0.1 | $p$=0.2 | $p$=0.3 |
| SFT | 66.0 | 76.7 | 83.3 | 84.8 | 24.8 | 22.4 | 22.0 | 21.4 |
| Lisa | **0** | **1.8** | 3.9 | 7.1 | 19.6 | 20.0 | 19.0 | 18.6 |
| DER | 0.5 | 2.3 | **3.5** | **4.8** | **22.0** | **20.8** | **20.0** | **20.6** |
| LwF | 2.8 | 2.7 | 55.6 | 59.9 | 20.4 | 20.4 | 18.2 | 19.6 |

Table 3: **Safety and Utility Under Different Harmful Ratios ($p$).** The table presents the impact of increasing the harmful ratio when fine-tuning the safety-aligned model LLaMA2-7B on the GSM8K dataset. DER appears most robust to increases in the harmful ratio compared to all reported approaches.

**Robustness to Harmful Ratio.** In Table 2, the harmful ratio was set to ($p = 0.1$). Here we investigate the impact of higher poison ratios ($p \in \{0.2, 0.3\}$) on DER, LwF, and Lisa. Table 3 shows the impact of increasing the harmful ratio on both ASR and utility on GSM8K. We can see that for SFT, that the higher the harmful ratio, the higher the ASR and the lower the utility. The decreasing utility is mainly because we fix the total number of fine-tuning samples. A higher poison ratio implies more harmful queries, and thus lower task samples.

DER shows strong robustness across different harmful ratios where the ASR reaches a maximum of 4.8% at $p = 0.3$ while maintaining utility $\geq 20$. Lisa shows moderate degradation, with ASR increasing to 7.1% and utility dropping to 18.6% at $p = 0.3$. In contrast, LwF suffers significant failure at higher poison ratios. Its safety collapses at $p = 0.2$ with ASR jumping to 55.6% and further increasing to 59.9% at $p = 0.3$. This aligns with prior work showing that knowledge distillation faces optimization difficulties when the student's training data significantly diverges from the teacher's training distribution Stanton et al. (2021), with harmful samples representing an extreme case of such distribution shift.

**Exaggerated Safety.** Model helpfulness involves not only producing answers that are truthful and aligned with human values, but also consistently answering safe prompts and avoiding unnecessary refusals. Safety alignment can induce exaggerated safety (over-rejection), where the model incorrectly refuses to answer safe prompts Arditi et al. (2024). We evaluate Lisa, DER, and LwF on three over-rejection benchmarks: XSTest Röttger et al. (2023), OR-Bench-80K, and OR-Bench-Hard Cui et al. (2024), which contain seemingly harmful but actually benign prompts. Table 4 reports refusal rates averaged over the three benchmarks. We adopt the open source model Wildguard[7] for refusal detection.

---

[7]https://huggingface.co/allenai/wildguard

| Method | GSM8K | | SST2 | | Code | |
|---|---|---|---|---|---|---|
| | ASR% ↓ | Refusal% ↓ | ASR% ↓ | Refusal% ↓ | ASR% ↓ | Refusal% ↓ |
| SFT | 76.7 | 1.3 | 78.3 | 1.0 | 86.9 | 0.9 |
| Lisa | **1.8** | 18.0 | 6.6 | 8.5 | **1.2** | 23.5 |
| DER | 2.3 | 14.4 | 5.9 | 8.9 | 1.7 | 16.1 |
| LwF | 2.7 | **3.0** | **2.1** | **1.4** | 4.3 | **2.1** |

Table 4: **Over-rejection vs. Safety.** We report safety (ASR) alongside over-rejection on safe prompts (Refusal) across three tasks. SFT shows very low refusal but is unsafe (high ASR). Among reported methods, LwF attains the best overall trade-off, low ASR with the lowest refusal rates, while DER and Lisa achieve similarly low ASR at the cost of higher refusal.

| | ASR% ↓ | | | Pass@1 ↑ | | |
|---|---|---|---|---|---|---|
| Methods | $m{=}50$ | $m{=}150$ | $m{=}250$ | $m{=}50$ | $m{=}150$ | $m{=}250$ |
| Lisa | 3.1 | 1.6 | 1.8 | 19.4 | 19.6 | 20.0 |
| DER | 3.5 | 2.4 | 2.3 | 21.8 | 20.4 | 20.8 |

Table 5: **Effect of Safety Sample Budget ($m$) on safety and utility.** Varying the number of safety samples used for fine-tuning LLaMA2-7B on GSM8K, using DER and Lisa, shows modest ASR increases at $m = 50$ and comparable ASR at $m = 150$ vs. $m = 250$ for both methods, with only small fluctuations in Pass@1.

SFT shows a very high ASR, which makes the model comply with most prompts, and that explains the low refusal rate ($\leq 1.3\%$). Among safety-preserving methods, LwF has the lowest refusal rates ($\leq 3\%$). DER and Lisa have higher refusal rates than LwF, likely because both methods use additional safety data during fine-tuning, which can increase sensitivity to potentially harmful patterns Bianchi et al. (2024). DER shows a moderately lower refusal rate ($\leq 16.1\%$) compared to Lisa ($\leq 23.5\%$) while being at comparable ASR.

**Effect of Safety Sample Budget.** Prior work shows that adding up to 5% safety data to downstream task data is sufficient to maintain safety behavior, while larger proportions can induce exaggerated safety Bianchi et al. (2024). In all presented results so far, we set the default number of safety samples to 5% of the downstream task data. That means, for GSM8K and SST2 the number of safety samples is $m = 250$, while for Code $m = 500$. We ablate the effect of reducing $m$ for Lisa and DER on GSM8K and present the results in Table 5. Overall, DER and Lisa show comparable ASR across the different number of safety samples $m$. As expected, the ASR increases slightly when we set $m = 50$ (compared to $m = 250$). Both approaches show slight fluctuations in utility across different safety sample budgets.

## 5.7 Summary and Genaralization

Our experiments demonstrate that continual learning approaches effectively preserve safety during fine-tuning. DER emerges as the most reliable method, achieving strong safety-utility balance across diverse scenarios. Generally, memory-based methods outperform regularization approaches in benign settings. Under poisoning, DER and LwF prove most effective. However, LwF fails catastrophically at high poison ratios, while DER maintains robustness even under severe poison ratios. The robustness of DER can be attributed to the combination of two properties: it operates in function space, directly constraining the model's output distribution rather than its parameters Pan et al. (2020); and its supervision signal consists of soft logit targets pre-computed from the aligned model and stored in a fixed buffer, encoding richer information than hard labels Hinton et al. (2015).

**Generalization Across Models.** Evaluation on Mistral-7B and Gemma-2B confirms findings generalize across architectures (Table 6). DER achieves best safety-utility trade-off on both models, maintaining strong

|  | Mistral | | Gemma | |
|---|---|---|---|---|
| Methods | ASR% ↓ | Pass@1 ↑ | ASR% ↓ | Pass@1 ↑ |
| SFT | 83.7 | 49.8 | 78.8 | 30.4 |
| Lisa | 19.5 | 48.4 | 0.6 | 21.0 |
| DER | 7.1 | 48.2 | 4.0 | 32.6 |
| LwF | 12.5 | 36.6 | 11.1 | 28.2 |

Table 6: **Generalization Across Model Architectures.** We evaluate DER, LwF, and LISA on GSM8K using Mistral-7B and Gemma-2B models. DER achieves the best safety-utility balance across both architectures.

safety while preserving or exceeding standard fine-tuning utility. These results establish CL approaches, particularly DER, as a robust solution for preserving safety alignment during fine-tuning.

## 6 Conclusion

Fine-tuning safety-aligned language models on downstream tasks poses a significant challenge: models can lose their safety alignment even when trained on benign data, with the problem becoming more severe when training data contains harmful content. In this work, we frame safety-preserving fine-tuning as a CL problem and investigate whether CL approaches can effectively mitigate safety degradation while maintaining utility on downstream tasks. We adapt and evaluate continual learning methods that span regularization-based (LwF, EWC), memory-based (A-GEM, DER, Refresh Learning), and model merging (MagMax) paradigms, and compare them to established baselines from the literature on three downstream tasks.

Our comprehensive evaluation reveals that CL methods effectively preserve safety alignment across both benign and poisoned fine-tuning scenarios. Among these approaches, DER achieves the optimal balance of safety and utility. It outperforms other CL methods and existing safety-preserving baselines, and remains robust even at high poison ratios. Our findings generalize across model architectures (LLaMA2-7B, Mistral-7B, Gemma-2B) and across tasks (GSM8K, SST2, Code) confirming the effectiveness of CL for safety-preserving fine-tuning.

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
