# OpenReview forum: "Unforgotten Safety: Preserving Safety Alignment of Large Language Models with Continual Learning"
_TMLR — Under review for TMLR_

### Review · Reviewer_Pa1o · 2026-06-04

**Summary Of Contributions:**

## Summary  of Contribution
This paper studies the safety alignment loss after fine-tuning LLMs on downstream tasks, and frames the problem as a continual learning (CL) challenge. The authors investigate whether existing CL techniques can preserve safety alignment while maintaining downstream task utility through experiments on both benign and poisoned fine-tuning settings. The results show that CL methods generally mitigate safety degradation compared to standard fine-tuning.

## Strengths
1. This paper is well written and easy to understand.
2. The experiments are comprehensive, covering multiple downstream tasks, model architectures, and both benign and poisoned fine-tuning settings.
3. Framing the safety alignment degradation problem as a CL problem is interesting.

## Weakness
1. The novelty of this paper is limited. The connection between safety alignment degradation and catastrophic forgetting has been discussed in [1]. Besides, existing safety-preserving fine-tuning approaches already employ mechanisms closely related to continual learning, such as replay or regularization.
2. The author didn't compare CL methods with safety-preserving methods on experiments on benign data. Additionally, the results in Table 2 show that the carefully designed safety-preserving methods are better than direct adaption of CL methods.
3. The poisoned setting mainly considers vanilla harmful samples. It remains unclear whether the conclusions would hold under more sophisticated attacks, such as covert malicious fine-tuning.
4. The paper mainly investigates existing CL methods, without introducing any new method. Given this limitation, the paper would benefit from deeper analysis explaining why certain CL methods (e.g., DER) are particularly effective for preserving safety alignment and what lessons can be drawn for future safety-preserving fine-tuning approaches.

## Reference

[1] Qi, X., Zeng, Y., Xie, T., Chen, P.-Y., Jia, R., Mittal, P., & Henderson, P. (2024).
*Fine-tuning Aligned Language Models Compromises Safety, Even When Users Do Not Intend To!*
International Conference on Learning Representations (ICLR 2024).

**Audience:**

Yes

**Audience Explanation:**

The paper investigates an important challenge in LLMs, which is safety alignment degradation after fine-tuning. The authors connect it with catastrophic forgetting, which may inspire some solutions.

**Broader Impact Concerns:**

None.

**Claims And Evidence:**

Yes

**Claims Explanation:**

The main claims of the paper are supported by the experimental evidence presented. The authors conduct evaluations across multiple downstream tasks, model families, and both benign and poisoned fine-tuning settings. The results consistently show that continual learning approaches can mitigate safety degradation compared to standard fine-tuning.

**Requested Changes:**

1. Add experiments on sophisticated attacks, such as covert malicious fine-tuning, to further investigate the effectiveness of CL methods.
2. The paper would benefit from deeper analysis explaining why certain CL methods (e.g., DER) are particularly effective for preserving safety alignment and what lessons can be drawn for future safety-preserving fine-tuning approaches.

---

### Review · Reviewer_Cq6b · 2026-06-14

**Summary Of Contributions:**

This paper studies safety degradation after specific fine-tuning. The main contributions include:
- adapt classical continual learning methods to the safety-preserving fine-tuning setup.
- empirically show that continual learning methods, especially DER, can serve as strong baselines for preserving safety alignment during benign and poisoned fine-tuning.

However, the algorithmic contribution is limited because the methods are largely direct adaptations of existing continual learning techniques rather than novel methodology.

**Audience:**

Yes

**Audience Explanation:**

This paper studies safety-preserving problem when users apply fine-tuning with their own data. This is an important problem because downstream fine-tuning can degrade previously learned safety behaviors, even when the base model has already undergone safety alignment. This issue is especially relevant to researchers and practitioners working on LLM customization, fine-tuning-as-a-service, and safe model deployment.

In this paper, a key finding is that continual learning methods, with proper adaptation, can contribute to safety-preserving. This is valuable to inform the audience of a possible approach to personalizing or specializing LLMs while keeping safe.

**Broader Impact Concerns:**

I don't see major ethical concerns, as the paper is mainly focused on improving the safety of fine-tuning LLMs. That said, the dual-use nature of studying poisoned fine-tuning should be marked. Also, the safety-preserving methods may increase over-refusal on benign prompts. These concerns do not undermine the value of the work, but they should be acknowledged when discussing implications.

**Claims And Evidence:**

Yes

**Claims Explanation:**

Overall, the main empirical claims are supported by reasonably clear evidence. For example, the paper claims that standard fine-tuning can degrade the safety alignment of LLMs, and the reported results show substantial increases in ASR after fine-tuning, especially under poisoned fine-tuning. The paper also claims that adapted continual learning methods can mitigate safety degradation, and this is directly supported by the benign and poisoned fine-tuning experiments, where several CL methods substantially reduce ASR compared with standard SFT while maintaining comparable downstream utility.

That said, some broader claims would benefit from stronger evidence. For example, the generalization claim across tasks and model families is only partially supported, since the cross-model experiments are conducted only on GSM8K. The robustness claim under poisoned fine-tuning would also be strengthened by using additional poison datasets or harmful data distributions.

**Requested Changes:**

## Paper Presentation
The authors are encouraged to pay more attention to presentation details. For example,
- the section 5.4.1 is "Fine-tuning on Benign Data". So there expects to be a "section **5.4.2** Fine-tuning on Poisoned Data". However, the "Fine-tuning on Poisoned Data" section is actually section **5.5**.
- In the tables, most numbers are rounded to one decimal place; even integers are written with a decimal point. However, in table 1, the number "22" for DER, GSM8K, Pass@1, should be "22.0". Similar typos happen in table 3.

## Methodology
The paper is mainly an empirical study of existing continual learning methods under the scenario of LLM fine-tuning. The paper would be significantly strengthened by either proposing a new CL-style method specifically designed for the safety-preserving fine-tuning problem, or by providing a deeper explanation of why the proposed adaptations are non-trivial and particularly suitable for LLM safety alignment.

## Experiments
1. The generalization experiments are only done on GSM8K, weakening the generalization claim. Additional cross-model experiments on SST2 and Code would make the generalization claim more convincing.
2. The main results in Tables 1 and 2 are based on LLaMA2-7B, while the additional model-family experiments are relatively limited. More experiments on recent open-source model families and different model scales, such as qwen3.5, gemma4, and llama3.x models, would better strengthen the generality of the findings.
3. One more poison dataset is expected to show robustness to different poison data.